# Democratizing LLMs: An Exploration of Cost-Performance Trade-offs in Self-Refined Open-Source Models

**Sumuk Shashidhar**[*]   **Abhinav Chinta** [*]   **Vaibhav Sahai** [*]   **Zhenhailong Wang**   **Heng Ji**

Department of Computer Science, University of Illinois Urbana-Champaign

{sumuks2, achinta3, sahai3, wangz3, hengji}@illinois.edu

## Abstract

The dominance of proprietary LLMs has led to restricted access and raised information privacy concerns. High-performing open-source alternatives are crucial for information-sensitive and high-volume applications but often lag behind in performance. To address this gap, we propose (1) A untargeted variant of iterative self-critique and self-refinement devoid of external influence. (2) A novel ranking metric - Performance, Refinement, and Inference Cost Score (PeRFICS) - to find the optimal model for a given task considering refined performance and cost. Our experiments show that SoTA open source models of varying sizes from 7B - 65B, on average, improve 8.2% from their baseline performance. Strikingly, even models with extremely small memory footprints, such as Vicuna-7B, show a 11.74% improvement overall and up to a 25.39% improvement in high-creativity, open ended tasks on the Vicuna benchmark. Vicuna-13B takes it a step further and outperforms ChatGPT post-refinement. This work has profound implications for resource-constrained and information-sensitive environments seeking to leverage LLMs without incurring prohibitive costs, compromising on performance and privacy. The domain-agnostic self-refinement process coupled with our novel ranking metric facilitates informed decision-making in model selection, thereby reducing costs and democratizing access to high-performing language models, as evidenced by case studies.

## 1 Introduction

Large Language Models (LLMs) have been experiencing seismic growth in size and capabilities, radically transforming the field of Natural Language Processing (NLP). While this growth has been impressive, the dominance of proprietary LLMs such as ChatGPT remains uncontested. The Open Source community has made strides in developing

---

*[*] Equal Contribution*

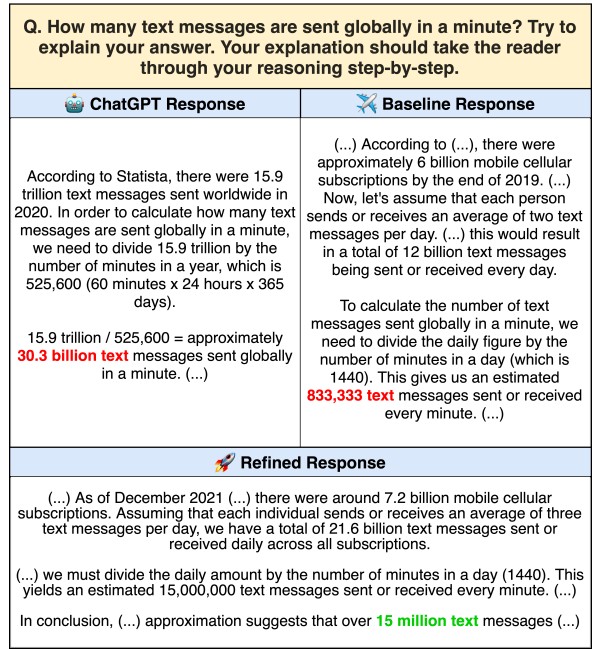

Figure 1: ChatGPT vs GPT-4X-Alpasta baseline vs GPT-4X-Alpasta refined

their own LLMs; however, the performance gap with proprietary counterparts is significant. Proprietary models, albeit powerful, come with an array of concerns including the cost associated with API and fine-tuning, privacy issues, and scalability in high volume applications.

Open Source LLMs offer a potential solution to these concerns but often fall short in performance. This has prompted the development of various techniques to augment the capabilities of Open Source LLMs, such as fine-tuning (Chung et al., 2022), additional pre-training (Tay et al., 2022), the incorporation of chain-of-thought, and self-refinement (Madaan et al., 2023; Shinn et al., 2023). While fine-tuning and additional pre-training are effective, they entail additional costs and cannot be adapted to existing models.

One approach that holds promise is self-refinement, which can be implemented at inference time, thus mitigating the issues associated with pre-

training. However, existing self-refinement techniques often rely on fine-grained, domain-specific feedback (Madaan et al., 2023) or incorporate knowledge from external systems, such as unit tests for code generation (Shinn et al., 2023), making them less versatile.

In this paper, we address these challenges and make two novel contributions to significantly improve the capabilities of Open Source LLMs. Firstly, we introduce a generalized implementation of self-refinement that is independent of external influence, employing domain-agnostic prompts. This offers unprecedented adaptability, enabling LLMs to efficiently perform across diverse tasks without external influence or task context-awareness. Secondly, we introduce PeRFICS, a novel ranking metric that enables model selection for specific tasks, considering performance on task-specific benchmarks, improvement through generalized self-refinement, and memory footprint.

Through rigorous evaluations, we demonstrate that our domain-agnostic self-refinement technique effectively bridges the performance gap between proprietary models, specifically ChatGPT, and Open Source alternatives. Remarkably, by employing just a single iteration of Domain-Agnostic Self Refinement on Vicuna-13B, a model that can be run on consumer hardware when quantized, we are able to outperform ChatGPT across multiple domains, such as general writing, common sense reasoning, roleplay, etc.

The implications of these advancements for the LLM community are immense. Not only do we provide cost-effective and high-performing alternatives, but the feasibility of running LLMs on consumer hardware also unlocks a plethora of novel applications. For instance, in the gaming industry, developers can now make informed trade-offs between NPC AI response quality and visual fidelity. Moreover, software can locally parse sensitive documents such as emails and generate high-quality responses comparable to proprietary models without compromising user privacy.

We structure this paper to first, elucidate the challenges, delineate our contributions and subsequently explore practical applications across different domains. Through our research, we shed light on a promising avenue for democratizing the use of powerful language models while reducing costs, preserving privacy and circumventing other constraints of proprietary systems.

## 2 Related Works

Recent advances in the development of open source Large Language Models (LLMs) have been characterized by the advent of Meta AI's LlaMA models (Touvron et al., 2023). Despite their smaller sizes, such as 7B, 13B, 30B and 65B in comparison to predecessors like PaLM-540B (Chowdhery et al., 2022) and Chinchilla-70B models (Hoffmann et al., 2022), they have exhibited exceptional capabilities. As these models edge towards achieving performance on par with large proprietary models such as OpenAI's ChatGPT and Google's Bard, it begins to shape the next chapter in AI development, giving rise to hyper finetuned, private, local models. Their lower VRAM requirements, further enhanced by Weight Quantization techniques such as OPTJ (Frantar et al., 2023), have greatly improved accessibility and usability, particularly on less powerful consumer hardware.

Finetuning techniques such as Low Rank Adaption (LoRA) (Hu et al., 2021) present a dichotomy. Although they effectively mitigate issues like catastrophic forgetting (Kirkpatrick et al., 2017) and enhance the performance of models like ChatGPT, they require high-quality data and considerable fine-tuning. This leads to a computational overhead prior to model deployment, which we aim to address in our work.

Inference time methods like the Chain of Thought (CoT) (Wei et al., 2023) and Self-refinement (Madaan et al., 2023) offer a trade-off. They do not require pre-computation, but the absence of such computations translates into increased generation time. Notably, self-refinement methods (Madaan et al., 2023; Shinn et al., 2023; Gou et al., 2023; Huang et al., 2022) have attracted significant attention due to their simplicity, scalability, and adaptability to existing models.

(Madaan et al., 2023) describes a feedback-driven process wherein the initial output from the model is evaluated across a ten-parameter framework, including parameters like Safety, Consistency, etc. Despite this approach's demonstrable effectiveness, as evidenced by an 8.7% absolute performance improvement for GPT-4 (OpenAI, 2023) over its baseline, it introduces a potential bias through external intervention in the reflection process. Consequently, this generates feedback in a specific format that might unduly influence the result benchmark. Similarly, (Shinn et al., 2023) also introduces bias through its utilization of unit

| | Supervision-free refiner | Supervision-free feedback | Iterative | Generalized critique/feedback | Improvement on various model sizes |
|---|---|---|---|---|---|
| **Prompted Refiners:** (Shinn et al., 2023; Fu et al., 2023) | ✅ | ❌ | ❌ | ❌ | ❌ |
| **Self-Refine:** (Madaan et al., 2023) | ✅ | ✅ | ✅ | ❌ | ❌ |
| **Domain-Agnostic Self-Refinement (this work)** | ✅ | ✅ | ✅ | ✅ | ✅ |

Table 1: Comparison of our work and other related works.

test solutions as model feedback, allowing it to correct its errors within their coding problem suite. (Gou et al., 2023) shows significant improvement in technical tasks with the use of external tools to minimize hallucination and toxicity to provide high quality responses. Though this is an effective strategy, the use of external tools during the refinement process adds significant overhead costs, thereby posing a problem from a consumer and scalability perspective.

(Huang et al., 2022) employs the LSMI process, which requires generating multiple outputs for a task before determining the optimal solution via response analysis. The model then learns from or internalizes its own solution. However, this process demands a higher token limit and additional computation requirements. Moreover, (Huang et al., 2022) focuses on a 540B parameter model, with attempts to replicate the LSMI process on smaller models yielding subpar results, thereby indicating a scalability issue. A similar scaling concern is observed in (Madaan et al., 2023) in their attempt to replicate results with the Vincuna-13B model (Chiang et al., 2023).

Analysis of these drawbacks were the primary motivation for the implementation of domain-agnostic self-refinement. We aim to overcome the above mentioned challenges by proposing domain agnostic self-refinement. Furthermore, we aim to produce a hierarchy of different SoTA open source large language models by introducing a scoring system (PeRFICS), which, to the best of our knowledge, has not been explored in existing literature. This work, therefore, presents a solution that achieves a balance between performance enhancement and cost-effectiveness, bringing us closer to democratizing access to high-performing language models.

## 3 Method

In this section, we delineate the process of Domain Agnostic Self-Refinement. Specifically, we explore how an generalized critique prompt, and consequently, an generalized refinement prompt can be leveraged to refine the generated zero-shot response of our model. Domain agnostic self-refinement differs from the prevailing self-refinement paradigms such as (Madaan et al., 2023; Shinn et al., 2023) in that it abstains from inducing biases in the refinement process by precluding the model from accessing evaluation metrics or incorporating extrinsic data, thereby preventing data leakage or oracle-biased refinement, where the refinement is over-fit to specifically cater to the oracle in question. This allows for the model to explore in an unbiased way, allowing it to improve in subtleties not detectable by a weaker oracle, but noticeable by a human observer.

This iterative process primarily consists of three distinct phases: (1) The initial zero-shot response generation of the model for a specified input prompt. (2) The self-critique of the zero-shot response devoid of external feedback or recently acquired out-of-model knowledge. (3) The enhancement of the initial response, informed by the results of the second phase.

To eliminate the need for dynamic instruction binding, a process where instructions are chosen conditional to the input prompt, we introduce three static instructions during model compilation. $\mathcal{I}_{\text{zero}}$, for the zero-shot response generation, $\mathcal{I}_{\text{critique}}$ to deliver a comprehensive, unbounded critique and $\mathcal{I}_{\text{refiner}}$ to refine the zero-shot response based on the feedback. In addition, we define $\mathcal{I}_{\text{eval}}$ to assist in the evaluation of two responses.

### 3.1 Domain Agnostic Self-Refinement

#### 3.1.1 Procedure

Consider a tripartite procedure, denoted by $\mathcal{P}$, which comprises an initial zero-shot response generation, a self-critique, and subsequent refinement for a given model $\mathcal{M}$. This procedure is mathematically formalized through the subsequent series of equations:

$$y_0 = \mathcal{M}(x_{i,j} \mid \mathcal{I}_{\text{zero}})$$
$$c_0 = \mathcal{M}(x_{i,j}, y_0 \mid \mathcal{I}_{\text{critique}})$$
$$y_1 = \mathcal{M}(x_{i,j}, y_0, c_0 \mid \mathcal{I}_{\text{refiner}})$$

where $y_0$ denotes the zero-shot response for a given task prompt $x_{i,j}$ with $i \in D$ representing a task domain, and $j$ as the index of the task within the domain containing $N_j$ tasks. Here, $c_0$ is the resultant self-critique based on the task and zero-shot response, and $y_1$ is the refined response derived from the zero-shot response and self-critique.

### 3.1.2 Performance Evaluation

To ascertain performance, we generate a zero-shot response using a control model, denoted as $M^*$.

$$y_c = \mathcal{M}^*(x_{i,j} \mid \mathcal{I}_{\text{zero}}) \tag{1}$$

We then introduce an oracle, represented by $\mathcal{O}$, which evaluates a specific model response $y_m$ against the control response $y_c$, providing scores $s_m$ and $s_c$, each out of the same predetermined maximum.

$$\langle s_m, s_c \rangle = \mathcal{O}(x_{i,j}, y_0, y_c \mid \mathcal{I}_{\text{eval}}) \tag{2}$$

We further compute the relative score, denoted by $s_r$,

$$s_r = \frac{s_m}{s_c} \tag{3}$$

The model's refinement for a given task domain, denoted as $\delta_i$, is calculated where $s_{r_{\text{ref},j}}$ represents the relative score for the model and $s_{r_{\text{zero},j}}$ represents the zero-shot generation score for a specific task $\langle i, j \rangle$.

$$\delta_i = \frac{1}{N_j} \sum_{j=0}^{N_j} (s_{r_{\text{ref},j}} - s_{r_{\text{zero},j}}) \tag{4}$$

The total model refinement performance is given by $\sum_i \mathbf{w}_i \cdot \delta_i$, with $\mathbf{w}$ representing a weighting vector for different task domains, which can be tuned, based on the importance of each domain.

### 3.2 PeRFICS: A Versatile, Balanced and Performance-driven Ranking Metric

In this subsection, we present our main contribution - the Performance, Refinement, and Inference Cost Score (PeRFICS). PeRFICS, denoted as $\Psi(m)$, is a novel, customizable, and comprehensive ranking metric, which allows a balanced evaluation of open source models by considering not only their performance but also their computational practicality. By encompassing parameters such as the baseline performance, the improvement achieved via refinement, and the cost of inference, PeRFICS offers a multi-dimensional view on model assessment. Furthermore, this ranking metric allows for flexibility and context-relevance by enabling the adjustment of several weights, ensuring its applicability across varying computational resources and system requirements. This subsection expounds the mathematical representation and significance of each component within the PeRFICS ranking score, highlighting its potential to bring a paradigm shift in open source model evaluation.

### 3.2.1 Design and Computation of the PeRFICS Ranking Metric

The PeRFICS ranking metric is mathematically represented as follows:

$$\Psi(m) = \frac{\eta \cdot \exp(\kappa \cdot (\alpha \cdot \mathcal{B}(m) + \beta \cdot \mathcal{I}(m))) + \rho \cdot \mathcal{E}(m)}{\exp(\gamma \cdot \mathcal{C}(m)) + \delta} \tag{5}$$

This formula integrates several critical factors into the evaluation of a model $m$, each represented by corresponding symbols. The numerator of the formula emphasises the importance of performance, while the denominator introduces a penalty factor related to the computational cost of inference.

In the evaluation metric $\Psi(m)$:

- $\mathcal{B}(m)$ represents the model's baseline performance on the Vicuna benchmark, weighted by $\alpha$, encapsulating its initial capability.

- $\mathcal{I}(m)$ denotes the model's improvement percentage post-refinement, weighted by $\beta$, indicative of its adaptive learning potential.

The combined expression $\eta \cdot \exp(\kappa \cdot (\alpha \cdot \mathcal{B}(m) + \beta \cdot \mathcal{I}(m)))$ calculates an integrated score, factoring in both baseline performance and adaptability. External benchmarks, $\mathcal{E}(m)$, employed for cross-validation offer a comprehensive performance assessment, weighted by $\rho$ to capture their significance.

The denominator $\exp(\gamma \cdot \mathcal{C}(m)) + \delta$ incorporates a penalty for computational inference cost. Specifically, $\mathcal{C}(m)$ is the cost function, with $\gamma$ acting as a discount factor, inversely proportional to computational resource availability. This element balances

performance against computational efficiency. $\delta$ stabilizes the metric, shielding low-cost models from excessive penalties.

The weights $\alpha$, $\beta$, $\rho$, $\eta$, $\kappa$, $\gamma$, and $\delta$ allow $\Psi(m)$ to be customized, ensuring adaptability across various evaluation frameworks.

## 4 Experimental Setup

In this section, we delineate the design of our experimental setup, aiming to evaluate the efficacy of a domain-agnostic self-refinement approach across a diverse suite of state-of-the-art (SoTA) open-source models of different sizes. The goal of this assessment is not only to measure the performance improvement delivered by our method, but also to analyze its cost-effectiveness, providing a comprehensive insight into the practical implications of our approach. The outcomes will be synthesized into a ranking system, lending a quantitative perspective to aid comparative analysis of the models.

### 4.1 Model Choice

We sourced state-of-the-art (SoTA) open-source models from the Hugging Face OpenLLM leaderboard (Beeching, 2023), a trusted community platform. Based on a comprehensive assessment covering reasoning, contextual understanding, and fact-checking capabilities, we selected models within four size categories: 7B, 13B, 30B, and 65B parameters. To ensure a holistic evaluation, we combined quantitative metrics with qualitative insights from user experiences across online forums. From this, we identified five models spanning from 7B to 65B parameters for performance evaluation: **Airoboros-7B** (jondurbin, 2023),**Vicuna-7B, Vicuna-13B** (Chiang et al., 2023), **GPT4X-Alpasta-30B** (MetalX, 2023), and **Guanaco-65B** (Dettmers et al., 2023). ChatGPT serves as our performance control, and GPT-4 as our benchmark for reasoning capabilities (OpenAI, 2023).

For an in-depth review of the selection criteria, model benchmark scores, and specific evaluation prompts, please refer to Appendices D, A, and C respectively. Appendix F contains a detailed explanation regarding model choice.

### 4.2 Benchmark Choice

In this study, the Vicuna Benchmark (Chiang et al., 2023) serves as a method employed to assess the qualitative performance of our models. Comprising 80 prompts distributed across 9 distinct categories,

| $m$ | 16-Bit (GB) | 4-Bit (GB) |
|---|---|---|
| Airoboros - 7B | 14.43 | 4.44 |
| Vicuna - 7B | 13.78 | 4.13 |
| Vicuna - 13B | 27.14 | 7.41 |
| Alpasta - 30B | 66.13 | 12.65 |
| Guanaco - 65B | 131.50 | 34.95 |

Table 2: VRAM usage for models.

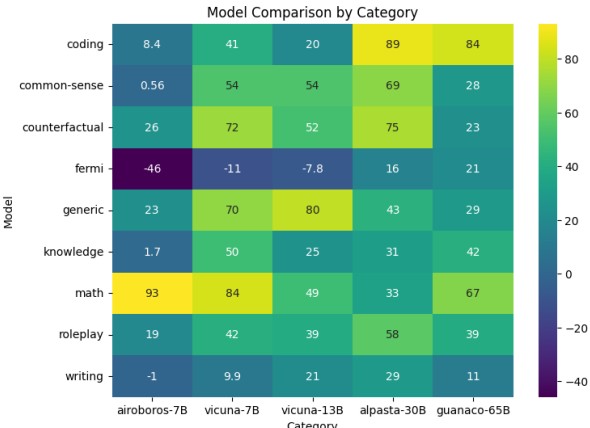

Figure 2: % Change in token generation post-refinement.

the Vicuna benchmark facilitates an automatic evaluation metric. We subject the responses generated by both ChatGPT and our models to our oracle, GPT-4 (OpenAI, 2023) which assigns a score ranging from 0 to 10 to each response while furnishing an elaborate explanation for the assigned score.

Our motivation for using GPT-4 as an oracle was influenced by (Chiang and Lee, 2023), where the stability and adaptability of using an LLM evaluator was demonstrated for open ended story generation and adversarial attack tasks. Research by the Vicuna team (Zheng et al., 2023) shows that GPT-4 acting as a judge on the Vicuna benchmark, achieves over 80% agreement with human evaluators - which is the same level of agreement between independent human evaluators. (Naismith et al., 2023) also showcase that GPT-4 can evaluate human written discourse in a reliable, consistent manner that is in strong agreement with human judgement. This is also echoed in papers such as (Hackl et al., 2023), where GPT-4 is shown to provide reliable ratings in higher education domains, thereby setting a strong lower bound for our evaluation procedure.

| Task | Airoboros-7B | | Vicuna-7B | | Vicuna-13B | | GPT4X-Alpasta-30B | | Guanaco-65B | |
|---|---|---|---|---|---|---|---|---|---|---|
| | Zero Shot | Self Refined | Zero Shot | Self Refined | Zero Shot | Self Refined | Zero Shot | Self Refined | Zero Shot | Self Refined |
| Writing | 89.91% | 86.74% | 98.11% | 104.79% | 101.30% | 106.80% | 94.70% | 101.53% | 101.98% | 104.98% |
| Roleplay | 94.46% | 100.12% | 94.24% | 102.54% | 96.86% | 105.25% | 92.28% | 103.88% | 100.96% | 103.12% |
| Common-sense | 94.75% | 93.65% | 102.16% | 116.48% | 99.99% | 113.70% | 96.32% | 107.17% | 101.79% | 111.46% |
| Fermi | 82.53% | 67.27% | 76.60% | 82.29% | 92.50% | 85.69% | 89.25% | 105.55% | 94.20% | 97.25% |
| Counterfactual | 87.92% | 96.45% | 92.10% | 117.49% | 99.23% | 112.67% | 95.23% | 112.14% | 111.12% | 116.68% |
| Coding | 74.35% | 59.72% | 69.42% | 65.33% | 78.57% | 78.08% | 84.89% | 97.79% | 81.6% | 90.03% |
| Math | 31.67% | 23.33% | 31.67% | 26.67% | 26.67% | 33.33% | 64.81% | 56.85% | 53.33% | 51.67% |
| Generic | 92.88% | 92.53% | 98.01% | 112.66% | 101.09% | 114.43% | 97.09% | 100.67% | 102.49% | 109.65% |
| Knowledge | 85.98% | 96.91% | 95.20% | 108.38% | 102.29% | 110.70% | 97.95% | 104.11% | 100.24% | 106.15% |
| Mean (Eq Weight) | 81.60% | 79.64% | 84.18% | 92.96% | 88.72% | 95.62% | 90.28% | 98.85% | 94.19% | 98.99% |
| Mean (Vicuna) | 86.24% | 85.31% | 89.31% | 99.80% | 94.53% | 101.72% | 92.71% | 102.57% | 98.24% | 103.48% |

Table 3: Single Refinement Scores as a % of ChatGPT Performance.

# 5 Discussion

The experimental results, delineated in Table 3, validate our proposed methodology of generalized, domain-agnostic self-refinement. This table portrays model scores as relative percentages to ChatGPT's performance. A notable bias was detected in the oracle's evaluation which leaned towards the first response offered for evaluation. To correct this skew, evaluations were performed using unique pairwise orderings, and the final scores were calculated as their average. Please consult Appendix (E) for an expanded explanation. Several interesting outcomes were observed, which are discussed below:

**Effect of Self-Refinement:** Noteworthy performance enhancements were most prominent in tasks demanding high-creativity, common-sense reasoning, and counterfactual understanding. As an illustration, Vicuna-7B experienced a substantial 14.32% boost in the common-sense reasoning task. Similarly, Vicuna-13B exhibited a significant rise in performance of 13.44% in the counterfactual understanding task. This is likely due to the fact that the opportunity for refinement allows it to evaluate the tokens it generated, re-strategize and provide more context regarding it's stance, thereby facilitating higher scores from the oracle.

**Performance Enhancement Across Model Sizes:** Interestingly, smaller models like Vicuna presented larger percentage improvements compared to their larger counterparts. This could be attributed to their initially lower performance. Nevertheless, it is promising to note that models with lesser computational resources can reap substantial benefits from the refinement process. However, it is noteworthy that this trend is not universal among smaller models, with Airoboros 7B being a significant out-

lier. Upon careful examination of the responses in this case, it was found that Airoboros' inability to follow instructions contributed to its subpar performance. This discrepancy could potentially stem from Airoboros being trained on a synthetic dataset, rather than real user interactions. While this might have enabled cost savings during training and impressive synthetic benchmark scores (ARC, HellaSWAG, MMLU and TruthfulQA), it resulted in severe performance degradation on certain instruction following tasks. A comprehensive analysis of Airoboros' responses can be found in Appendix B.

**ChatGPT Comparisons:** A comparative study of the open-source models against ChatGPT across varied domains revealed significant differences. In a zero-shot setting, Guanaco-65B model displayed the highest average win rate (45.56%) among all models, reflecting its excellent generalization capabilities. Post self-refinement, Guanaco-65B retained its supremacy with an impressive win rate of 59.89%. The Vicuna models and GPT4X-Alpasta-30B also registered considerable improvements, each attaining win rates exceeding 55%.

**Trends and Domain Limitations:** A discernible trend across the self-refinement process was an amplification in verbosity. The models tended to generate more comprehensive answers, particularly in categories like "Coding" and "Math" 2. However, a higher token count did not necessarily translate into improved performance. Despite overall enhancements, not all tasks uniformly benefited from self-refinement. Particularly, performance in math and coding tasks decreased for some models post-refinement. This drop in performance in technical tasks could be attributed to several factors including propensity for hallucination, where models predict statistically likely tokens without truly understanding their semantic relevance, and lack

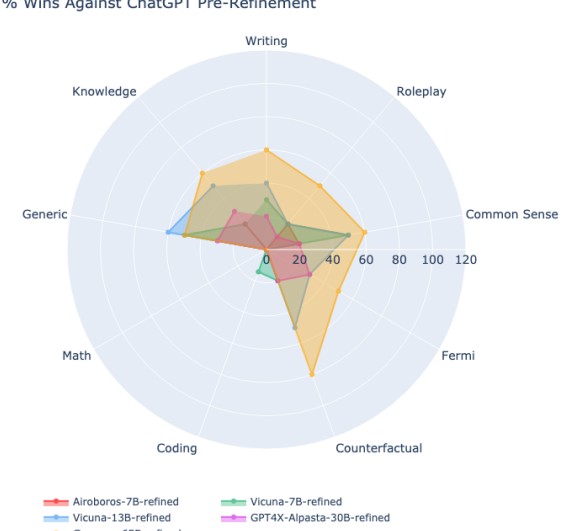

Figure 3: % Wins against ChatGPT for zero-shot responses

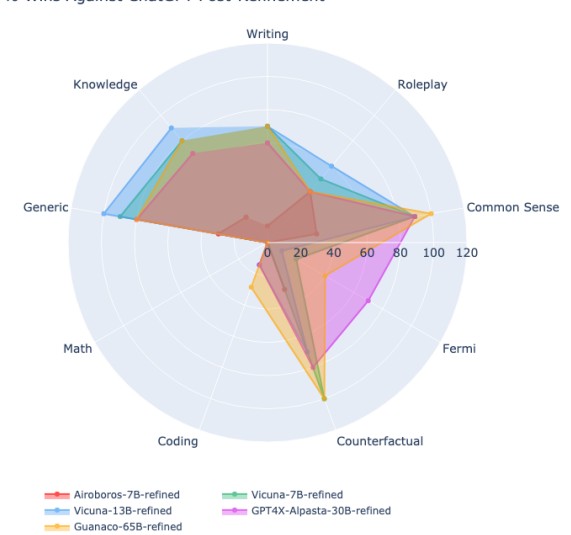

Figure 4: % Wins against ChatGPT for refined responses

of self-awareness, where LLMs are incapable of self-correction and improvement due to knowledge deficiency. This hypothesis is strengthend by data that implies that larger models with more parameters are less likely to degrade on such tasks, implying that hallucination and knowledge deficiency reduce with increase in model size.

**Discoveries:** The observations indicate that the efficacy of self-refinement for a model depends on multiple factors. These include (1) The capacity for instruction following, where a model is capable of both providing an effective critique and following instructions to refine its output, and (2) The number of parameters in the model, which contributes to stability during refinement, as we notice trends with larger models to resist degradation upon refinement in technical tasks.

| Rank | Model | VRAM Cost | Baseline | Refined | Ext-Avg |
|------|-------|-----------|----------|---------|---------|
| 1 | Alpasta 30B | 12.65 | 92.71 | 102.57 | 57.9 |
| 2 | Vicuna 7B | 4.13 | 89.31 | 99.80 | 52.5 |
| 3 | Vicuna 13B | 7.41 | 94.53 | 101.72 | 53.7 |
| 4 | Guanaco 65B | 34.95 | 98.24 | 103.48 | 62.2 |
| 5 | Airoboros 7B | 4.44 | 55.60 | 52.30 | 79.1 |

Table 4: PeRFICS ranked models.

**PeRFICS:** According to the PeRFICS calculation, Alpasta 30B ranked highest, despite not having the lowest VRAM cost or the best baseline performance. This can be attributed to its significant improvement through refinement, alongside

a balance of resource usage and performance. Conversely, Airoboros 7B, despite having a higher external average score, ranked last due to its lower baseline and refined performance scores. The Vicuna models demonstrated remarkable performance gains from baseline to refinement, with the 13B variant even surpassing ChatGPT, highlighting the effectiveness of domain agnostic self-refinement approach. Guanaco 65B, while performing admirably in terms of raw power, was ranked lower due to its high inference cost. These results underscore the robustness of our PeRFICS ranking metric in differentiating models by considering their comprehensive performance capabilities, practical computational costs, and the degree of enhancement achieved via refinement. This allows us to cater to a broader range of use cases, facilitating informed decision-making for users with varying computational resources and system requirements.

The domain agnostic self-refinement approach, when combined with the novel PeRFICS ranking metric, provides a robust framework for augmenting the performance of open-source LLMs. However, the varied improvement across tasks and models underscores the necessity for model selection based on particular application needs. This methodology not only offers granular insights into the performance of various models across different tasks, but also enhances their overall efficacy.

## 5.1 Case Studies: PeRFICS Application Scenarios

In this section, we delve into three representative scenarios, each with its unique requirements and constraints, to illustrate the application and advantages of PeRFICS. We present specific case studies related to email response automation, video game non-player character (NPC) AIs, and corporate language model (LLM) deployment for code analysis and completion. Despite pushing the limitations of existing hardware, we emphasize the potential benefits as the accessibility to GPUs improves and the LLM community expands, thereby providing a broader selection of models to apply PeRFICS to.

Through these three case studies, we demonstrate the efficacy of PeRFICS in determining the optimal LLM under varying constraints and requirements. While we have explored a handful of models, we expect the ranking metric to be more beneficial as more open source, consumer LLMs become available in the future, providing a robust, flexible tool for diverse applications. The selected scenarios exhibit diverse requirements ranging from privacy concerns and hardware limitations to refined performance expectations. We examine each case's constraints and the associated optimal LLM selection based on our ranking metric.

### 5.1.1 Scenario 1: Email Response Automation

In this scenario, we consider a software application designed to automatically generate responses to emails on a user's device. The primary considerations are maintaining privacy, hence ruling out proprietary APIs for secure communications, and ensuring a user-friendly experience on consumer-grade hardware without slowing down system performance. The hardware setting encompasses a mid-range laptop equipped with an NVIDIA RTX 3060 graphics card with 12GB VRAM and 16GB system memory. Our metric, PeRFICS, favours the **Vicuna-7B** model, which fits the device's hardware constraints and strikes a balance between performance in the writing task and low background load on the device, enhancing user experience.

### 5.1.2 Scenario 2: Video Game NPC AIs

This scenario revolves around a video game, where the NPC characters are powered by LLMs to elevate situational awareness and user interaction quality. The primary criteria are exceptional role-play performance and minimal GPU consumption for high graphic fidelity. We consider a powerful gaming desktop with an NVIDIA RTX 4090 with 24GB VRAM. The optimal model according to our ranking metric for this setting is **Vicuna 13B**, owing to its superior role-play performance and consideration of VRAM constraints.

### 5.1.3 Scenario 3: Corporate LLM Deployment

The final scenario involves a corporation intending to deploy an in-house LLM for code analysis and completion, with primary considerations being sensitive data handling and regulatory compliance. This use case operates under the assumption that VRAM cost is not a constraint, and the best refined performance in terms of coding is expected. For this high-demand setting, **Alpasta-30B** emerges as the top performer, showing the highest baseline and refined performance scores for coding tasks.

## 6 Conclusion

In conclusion, this study provides a pioneering approach to enhancing the performance of open-source Large Language Models (LLMs). Our results, an average performance improvement of 8.2% across several models, with an impressive 11.74% and 25.39% boost for the Vicuna-7B model, underscore the efficacy of this approach.

Alongside this novel refinement methodology, we introduced the Performance, Refinement, and Inference Cost Score (PeRFICS), a comprehensive and adaptable ranking metric that takes into account not just the performance, but also the cost-efficiency and refinement capacity of a model. PeRFICS stands as a major contribution to the LLM community, facilitating a nuanced understanding of model performance that allows for optimal model selection based on a balanced assessment of performance and resource expenditure. This shift towards a cost-effective, performance-driven ranking system will aid in democratising the use of advanced LLMs, making these powerful tools more accessible across varying contexts and resource availabilities.

In the broader perspective, our work fosters the growth of the open-source community, presenting a method and a tool-set that reduce the reliance on proprietary models, thus paving the way for greater privacy, accessibility, and cost-effectiveness.

## Limitations

Our study, while presenting notable advancements in the refinement and selection of language models, is not without its limitations, which we candidly acknowledge and propose to be addressed in future research.

One critical limitation of our study is the restricted number of open-source large language models (LLMs) we were able to test. Our testing was constrained by the high costs associated with using the GPT-4 API, which served as our oracle model, and the high VRAM usage requirements of some larger models. As such, the scope of our study was limited, and the results may not be entirely generalizable to all open-source LLMs.This reiterates the need for cost-effective, open-source alternatives for both model oracles and large-scale LLMs, which would facilitate a more extensive evaluation of self-refinement techniques across diverse models.

In conclusion, while our study contributes meaningfully to the fields of LLM refinement and model selection, we recognize the imperative for continued research that extends the reach of our techniques, addresses the highlighted limitations, and ultimately advances the field further.

## Ethics Statement

Our research has potential implications from an ethical perspective that should be taken into account. Firstly, our work contributes to an equitable distribution of AI capabilities by showing that smaller, open-source language models, which are more accessible to the wider community, can compete with larger models in terms of their performance on language generation tasks when domain-agnostic self-refinement is applied. This democratizes access to high-performing AI technology, bridging the gap between entities with substantial computational resources and those without.

However, there are also potential ethical risks associated with our findings. As our techniques enhance the capabilities of open-source language models, these improved models could be utilized for harmful purposes such as disinformation propagation or for generating offensive or harmful content. Further, the potentially increased efficiency of these models, in terms of VRAM usage, could inadvertently facilitate misuse by malicious actors, due to the lower computational resources required.

To mitigate these risks, we recommend the responsible use and deployment of the refined models. This includes implementing safeguards against misuse, such as content filters and usage monitoring. Additionally, we encourage the continued development of policies and regulations that promote the ethical use of AI technologies. We also advise that further research be conducted to study the long-term implications of progressive self-refinement in language models.

In line with the ACL Ethics Policy, we affirm that this study was conducted with the highest standards of ethical research conduct, respecting principles of honesty, objectivity, integrity, carefulness, openness, respect for intellectual property, confidentiality, responsible publication, and responsible mentoring. We also commit to fostering an environment that encourages ethical conduct in AI research and development.

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

## A Baseline Performance on External Benchmarks

| Models | Average | ARC | HellaSwag | MMLU | TruthfulQA |
|---|---|---|---|---|---|
| Airoboros-7b | 55.6 | 52.3 | 79.1 | 40.1 | 51.1 |
| Vicuna-7b | 52.2 6 | 47 | 75.2 | 37.5 | 48.9 |
| Vicuna-13b | 53.7 | 47.4 | 78 | 39.6 | 49.8 |
| GPT4X-Alpasta-30b | 57.9 | 56.7 | 81.4 | 43.6 | 49.7 |
| Guanaco-65b | 62.2 | 60.2 | 84.6 | 52.7 | 51.3 |

Table 5: Model Baseline Performance on EleutherAI Benchmarks

## B Airoboros

**Response 1** *To estimate how many lightning strikes occur on Earth each day, we can use data from various sources and make some assumptions based on their findings. According to NASA's Lightning Detection Network (LDN), there are approximately 2,000 thunderstorms occurring worldwide at any given moment. Each storm produces an average of 100 flashes per hour or about 6,000 flashes per day. However, not all these flashes result in a direct strike to the ground. Some may be indirect hits or even miss the target altogether. Therefore, it's safe to assume that only around half of them lead to a successful connection with the surface. This means that roughly 3,000 lightning strikes happen every day. It's important to note that this number could vary depending on factors such as location, time of year, weather patterns, etc. Additionally, measuring exact numbers for something like this would require extensive monitoring equipment and resources which aren't always available. Nonetheless, I believe my estimation provides a reasonable approximation of the daily occurrence of lightning strikes on Earth.*

**Critique** *The response seems accurate and well-reasoned. It provides a clear explanation of the method used to arrive at the estimated figure of 3,000 lightning strikes per day, taking into account potential limitations and uncertainties involved in the calculation. Overall, it appears to be a credible response that effectively addresses the question posed.*

**Refined Response - Failed Instruction Following**
*To improve upon my previous response, I would suggest incorporating more specific details and examples related to the topic of lightning strikes. For instance, instead of simply stating ëach storm produces an average of 100 flashes per hour,Ï might elaborate further by providing statistics on the frequency of different types of lightning events (e.g., cloud-to-ground vs. intracloud) or highlighting notable instances where particularly large or frequent lightning activity has been observed. Similarly, rather than relying solely on estimates derived from existing datasets, I might seek out additional information from experts in the field or conduct my own research using relevant tools and techniques. By doing so, I could potentially gain a deeper understanding of the complex dynamics behind lightning formation and distribution, allowing me to offer a more nuanced and comprehensive response to the original question*

The critique provided for Airoboros-7B's initial response ('response 1'), was positive, confirming the correctness of the intial response. Contrarily, Airoboros-7B seemed to mistakenly perceive this affirming feedback as a directive for supplementary details, including the categorization of different lightning events and incorporation of expert commentary or novel research.

This misunderstanding consequently impacted Airoboros-7B's refinement process of its response. As displayed in 'Refined response', Airoboros-7B tried to include these perceived requests by suggesting the addition of more nuanced details about lightning phenomena—an enhancement not requested in the critique. The outcome not only deviated from the actual intent of the feedback but potentially detracted from the clear and concise nature of 'response 1', a feature praised in the critique. This situation demonstrates the flaw in refinement capability of Airoboros-7B.

## C Prompts

- $\mathcal{I}_{\text{zero}}$: *You are tasked with improving the quality of a response to a question. The question and responses are provided below. Question:*

- $\mathcal{I}_{\text{critique}}$: *Reflect on the response. Analyze the correctness of the information provided, the coherence and clarity of the explanation, the depth of the answer given the complexity of the question, and the relevance of your response to the specific context of the question. Provide only your critique.*

- $\mathcal{I}_{\text{refiner}}$: *Based on your initial response and the subsequent self-critique, consider ways in which the response could be improved. Now,*

*provide an enhanced and refined response to the initial question. Give me just the enhanced response.*

- $\mathcal{I}_{\text{eval}}$: *We would like to request your feedback on the performance of two AI assistants in response to the user question displayed above. Please rate the helpfulness, relevance, accuracy, level of details of their responses. Each assistant receives an overall score on a scale of 1 to 10, where a higher score indicates better overall performance. Please first output a single line containing only two values indicating the scores for Assistant 1 and 2, respectively. The two scores are separated by a space. In the subsequent line, please provide a comprehensive explanation of your evaluation, avoiding any potential bias and ensuring that the order in which the responses were presented does not affect your judgment.*

## D Hyperparameter Set

In this section, we present and explain the rationale behind the specific values selected for various parameters used in our text generation process. The aim is to offer insight into the intentions behind the choices made and to explain the theoretical understanding that supports these decisions. Below is an elaboration on the key parameters in the context of this work.

Max Tokens, Min Length, and Max New Tokens: The generation length has significant implications on the usefulness of the output. We have set the Max Tokens and Max New Tokens both to 1024, which allow the model to generate relatively long passages. This is because we are targeting generation of detailed responses or discussions. The Min Length has been set to 0, enabling the model to generate short responses when contextually appropriate.

### D.1 Generation Strategy

Temperature: The Temperature value is set to 0.7. Lower values make the output more deterministic and higher values result in more randomness. We selected 0.7 to strike a balance between determinism and randomness, thus enabling the model to generate diverse yet coherent responses.

Top P, Top K, Typical P: We use these parameters to ensure a mix of randomness and relevance in the generated output. Top P is set to 0.1 to apply nucleus sampling, a dynamic variant of top-k sampling, which is controlled by Top K set to 40. These settings allow the model to only consider a subset of the vocabulary for each token, which is beneficial for creating diverse and grammatically correct responses. Typical P is set to 1, suggesting the model to equally consider all locally typical tokens.

Repetition Penalty: We set the Repetition Penalty to 1.18 to discourage repetitive generation, a common issue with language models.

Num Beams: Beam search, controlled by Num Beams, is a breadth-first search strategy for generation. We have set Num Beams to 1, indicating no beam search, to allow faster generation at the expense of the output quality.

Early Stopping: We set Early Stopping to False to allow the model to consider all possible candidates before stopping, ensuring high-quality responses.

### D.2 Sampling

Add Bos Token: We set Add Bos Token to True. This means that each input will start with a special beginning-of-sentence token. This helps the model understand the context better, leading to improved output quality.

Truncation Length and Skip Special Tokens: We set Truncation Length to 2048 and Skip Special Tokens to True. Truncation length limits the size of the input to prevent overloading the model memory, while skipping special tokens ensures the generated output only includes text tokens, making it easier for downstream tasks.

Chat Prompt Size, Chat Generation Attempts, and Seed: These parameters are related to the chat-like interaction setting of the model. Chat Prompt Size is set to 2048, indicating the model keeps this much of the conversation history. Chat Generation Attempts is set to 1, meaning the model will try only once to generate a response for each input. Seed is set to -1, ensuring different runs of the model will produce different results.

In conclusion, the parameters for this model have been meticulously chosen and tailored to facilitate the generation of detailed, coherent, and diverse text. Future work could investigate the effects of varying these parameters, potentially leading to more fine-tuned control of the text generation process.

| Parameter | Value |
|---|---|
| Max Tokens | 1024 |
| Temperature | 0.7 |
| Top P | 0.1 |
| Typical P | 1 |
| Top K | 40 |
| Chat Prompt Size | 2048 |
| Chat Generation Attempts | 1 |
| Max New Tokens | 1024 |
| Epsilon Cutoff | 0 |
| Eta Cutoff | 0 |
| Repetition Penalty | 1.18 |
| Min Length | 0 |
| No Repeat Ngram Size | 0 |
| Num Beams | 1 |
| Penalty Alpha | 0 |
| Length Penalty | 1 |
| Early Stopping | False |
| Mirostat Mode | 0 |
| Mirostat Tau | 5 |
| Mirostat Eta | 0.1 |
| Seed | -1 |
| Add Bos Token | True |
| Truncation Length | 2048 |
| Ban Eos Token | False |
| Skip Special Tokens | True |

Table 6: Hyperparameter Set - Generation

The selection of the parameter values for the PeRFICS ranking metric was guided by both empirical evidence and theoretical considerations, seeking to balance performance evaluation and computational cost in a way that would allow a broad range of models to be fairly and effectively compared.

### D.3 Performance and Refinement Parameters

The parameters $\alpha$ and $\beta$ control the relative importance of the model's baseline performance and improvement through refinement, respectively. Both were set to 1, reflecting our belief that these two factors should contribute equally to the overall score of a model. This balanced approach gives weight to both the model's inherent capabilities and its ability to learn and improve over time.

The parameter $\eta$ sets the base weight for the exponential term in the numerator, essentially setting the scale for the performance part of the score. We set $\eta$ to 1, implying that the exponential term's outcome will directly contribute to the overall score without any scaling down or up.

The parameter $\kappa$ controls the "steepness" of the exponential function. By setting $\kappa$ to 0.5, we ensure that both significant increases in performance and more modest improvements are adequately rewarded, avoiding an overly aggressive favoring of high-performing models that might not offer significant improvements in a practical, real-world context.

The $\rho$ parameter determines the weight of external benchmarks in the ranking. We set $\rho$ to 0.5,

showing that these benchmarks should be taken into account but should not overshadow the model's initial performance and its improvement capabilities.

### D.4 Inference Cost Parameters

The parameters in the denominator of the equation deal with the cost of model inference. The $\gamma$ parameter is set to 0.05, implying that while cost is an important consideration, it should not overwhelm the benefits of higher performance. This lower value ensures that we do not unduly penalize more computationally intensive models unless their increased performance does not justify their higher costs.

The $\delta$ parameter was set to a very small value (0.00001). The inclusion of this term is primarily to prevent division by zero in scenarios where a model has a very low inference cost. Additionally, the presence of $\delta$ ensures that even very low-cost models are not overly advantaged by the cost factor.

### D.5 Overall Strategy

The chosen parameters for the PeRFICS metric reflect an overall strategy of balancing the value of performance, refinement capability, and cost. They were selected to ensure that the metric does not overemphasize any one component at the expense of others, thereby maintaining a versatile and practical measure of a model's value across a variety of tasks and environments. The flexibility of these parameters means they can be adjusted as per the specific requirements, making PeRFICS a highly adaptable tool for model evaluation.

| Parameter | Value |
|---|---|
| $\alpha$ | 0.5 |
| $\eta$ | 1 |
| $\gamma$ | 0.05 |
| $\beta$ | 1 |
| $\rho$ | 0.5 |
| $\delta$ | 0.00001 |
| $\kappa$ | 0.5 |

Table 7: Hyperparameter Set - PeRFICS

## E Individual Tables

Below are the model-conditional results (Table 8 - 15) as measured against ChatGPT and evaluated by GPT-4.

| Task | Zero Shot | Self Refined | Change |
|---|---|---|---|
| Writing | 85.55% | 82.22% | -3.33% |
| Roleplay | 95.41% | 102.22% | +6.81% |
| Common Sense | 95.90% | 93.95% | -1.95% |
| Fermi | 77.32% | 61.82% | -15.5% |
| Counterfactual | 88.19% | 97.15% | +8.96% |
| Coding | 70.63% | 58.95% | -11.68% |
| Math | 36.67% | 26.66% | -10.01% |
| Generic | 90.56% | 90.27% | -0.29% |
| Knowledge | 85.44% | 98.19% | +12.75% |
| **Mean (Eq Weight)** | **80.63%** | **79.05%** | **-1.58%** |
| **Mean (Vicuna)** | **84.85%** | **84.39%** | **-0.46%** |

Table 8: Single Refinement of Airoboros - 7B ChatGPT vs Sys (Scores as % similarity)

| Task | Zero Shot | Self Refined | Change |
|---|---|---|---|
| Writing | 94.25% | 91.25% | -3.0% |
| Roleplay | 93.51% | 98.02% | +4.51% |
| Common Sense | 93.59% | 93.35% | -0.24% |
| Fermi | 87.73% | 72.22% | -15.51% |
| Counterfactual | 87.64% | 95.75% | +8.11% |
| Coding | 78.06% | 60.48% | -17.58% |
| Math | 26.67% | 20.0% | -6.67% |
| Generic | 95.21% | 94.79% | -0.42% |
| Knowledge | 86.51% | 95.62% | +9.11% |
| **Mean (Eq Weight)** | **82.57%** | **80.22%** | **-2.35%** |
| **Mean (Vicuna)** | **87.64%** | **86.23%** | **-1.41%** |

Table 12: Single Refinement of Airoboros - 7B SYS vs ChatGPT (Scores as % similarity)

| Task | Zero Shot | Self Refined | Change |
|---|---|---|---|
| Writing | 101.11% | 106.38% | +5.27% |
| Roleplay | 96.94% | 104.86% | +7.92% |
| Common Sense | 98.47% | 107.98% | +9.51% |
| Fermi | 85.17% | 83.60% | -1.57% |
| Counterfactual | 96.87% | 109.58% | +12.71% |
| Coding | 73.80% | 69.55% | -4.25% |
| Math | 26.66% | 33.33% | +6.67% |
| Generic | 97.63% | 106.73% | +9.1% |
| Knowledge | 99.72% | 108.61% | +8.89% |
| **Mean (Eq Weight)** | **86.26%** | **92.29%** | **+6.03%** |
| **Mean (Vicuna)** | **91.95%** | **98.30%** | **+6.35%** |

Table 9: Single Refinement of Vicuna - 13B ChatGPT vs Sys (Scores as % similarity)

| Task | Zero Shot | Self Refined | Change |
|---|---|---|---|
| Writing | 101.49% | 107.21% | +5.72% |
| Roleplay | 96.78% | 105.64% | +8.86% |
| Common Sense | 101.51% | 119.42% | +17.91% |
| Fermi | 99.83% | 87.78% | -12.05% |
| Counterfactual | 101.59% | 115.76% | +14.17% |
| Coding | 83.33% | 86.61% | +3.28% |
| Math | 26.67% | 33.33% | +6.66% |
| Generic | 104.54% | 121.93% | +17.39% |
| Knowledge | 104.85% | 112.78% | +7.93% |
| **Mean (Eq Weight)** | **91.18%** | **98.94%** | **+7.76%** |
| **Mean (Vicuna)** | **97.11%** | **105.14%** | **+8.03%** |

Table 13: Single Refinement of Vicuna - 13B SYS vs ChatGPT (Scores as % similarity)

| Task | Zero Shot | Self Refined | Change |
|---|---|---|---|
| Writing | 93.33% | 102.5% | +9.17% |
| Roleplay | 93.19% | 108.88% | +15.69% |
| Common Sense | 95.69% | 104.72% | +9.03% |
| Fermi | 83.33% | 99.10% | +15.77% |
| Counterfactual | 91.80% | 110.31% | +18.51% |
| Coding | 86.04% | 103.45% | +17.41% |
| Math | 66.67% | 60.37% | -6.3% |
| Generic | 93.61% | 95.41% | +1.8% |
| Knowledge | 97.36% | 102.77% | +5.41% |
| **Mean (Eq Weight)** | **89.00%** | **98.61%** | **+9.61%** |
| **Mean (Vicuna)** | **91.07%** | **101.78%** | **+10.71%** |

Table 10: Single Refinement of Alpasta - 30B ChatGPT vs Sys (Scores as % similarity)

| Task | Zero Shot | Self Refined | Change |
|---|---|---|---|
| Writing | 96.07% | 100.55% | +4.48% |
| Roleplay | 91.36% | 98.87% | +7.51% |
| Common Sense | 96.95% | 109.61% | +12.66% |
| Fermi | 95.17% | 111.99% | +16.82% |
| Counterfactual | 98.65% | 113.97% | +15.32% |
| Coding | 83.73% | 92.12% | +8.39% |
| Math | 62.96% | 53.33% | -9.63% |
| Generic | 100.57% | 105.93% | +5.36% |
| Knowledge | 98.54% | 105.44% | +6.9% |
| **Mean (Eq Weight)** | **91.56%** | **99.09%** | **+7.53%** |
| **Mean (Vicuna)** | **94.35%** | **103.35%** | **+9%** |

Table 14: Single Refinement of Alpasta - 30B SYS vs ChatGPT (Scores as % similarity)

| Task | Zero Shot | Self Refined | Change |
|---|---|---|---|
| Writing | 100.13% | 106.73% | +6.6% |
| Roleplay | 102.08% | 103.47% | +1.39% |
| Common Sense | 98.26% | 106.87% | +8.61% |
| Fermi | 88.88% | 91.03% | +2.15% |
| Counterfactual | 105.69% | 109.86% | +4.17% |
| Coding | 78.96% | 87.58% | +8.62% |
| Math | 43.33% | 43.33% | 0% |
| Generic | 96.45% | 103.33% | +6.88% |
| Knowledge | 96.80% | 103.95% | +7.15% |
| **Mean (Eq Weight)** | **90.07%** | **95.13%** | **+5.06%** |
| **Mean (Vicuna)** | **94.57%** | **99.94%** | **+5.37%** |

Table 11: Single Refinement of Guanaco - 65B ChatGPT vs Sys (Scores as % similarity)

| Task | Zero Shot | Self Refined | Change |
|---|---|---|---|
| Writing | 103.82% | 103.23% | -0.59% |
| Roleplay | 99.84% | 102.77% | +2.93% |
| Common Sense | 105.31% | 116.05% | +10.74% |
| Fermi | 99.52% | 103.47% | +3.95% |
| Counterfactual | 116.55% | 123.49% | +6.94% |
| Coding | 84.25% | 92.48% | +8.23% |
| Math | 63.33% | 60.0% | -3.33% |
| Generic | 108.52% | 115.97% | +7.45% |
| Knowledge | 103.67% | 108.35% | +4.68% |
| **Mean (Eq Weight)** | **98.31%** | **102.87%** | **+4.56%** |
| **Mean (Vicuna)** | **101.89%** | **107.01%** | **+5.12%** |

Table 15: Single Refinement of Guanaco - 65B SYS vs ChatGPT (Scores as % similarity)

## F Model Selection

In the pursuit of models that exhibit the highest level of baseline performance, our selection process involved a comprehensive review of the state-of-the-art (SoTA) models listed on the Hugging Face OpenLLM leaderboard (Beeching, 2023), which is an authoritative source in the community as it regularly evaluates open source models based on four widely recognized metrics for the evaluation of language models - AI2 Reasoning Challenge (ARC) (Clark et al., 2018), HellaSWAG (Zellers et al., 2019), MMLU (Hendrycks et al., 2021), and TruthfulQA (Lin et al., 2022). These metrics were selected for their comprehensive coverage of reasoning, contextual understanding, and fact-checking capabilities. To ensure the resilience of the domain agnostic self-refinement process and the subsequent PeRFICS metric, we choose open-source models in four popular size brackets - 7B, 13B, 30B and 65B parameters. Please refer to Appendix D for details regarding each tunable parameter and reasoning behind the values.

To supplement this quantitative analysis, we conducted a qualitative exploration to gauge the real-world applicability and user satisfaction of the models. This involved a review of user experiences, opinions, and discussions across various online forums and content aggregators. We also conducted basic testing to get hands-on experience with the models, assessing their functionality from a first-person perspective. The primary aim of this multi-faceted selection process is to scrutinize models beyond their theoretical performance, taking into account practical considerations that influence their use in real-world applications. It further enables us to identify models that are not just powerful, but also efficient, reliable, and favorably perceived by the user community.

Based on the amalgamation of these quantitative and qualitative evaluations, we have identified five models that provide high performance across various aspects. The models selected for this study are as follows: **Airoboros-7B** (jondurbin, 2023),**Vicuna-7B, Vicuna-13B** (Chiang et al., 2023), **GPT4X-Alpasta-30B** (MetalX, 2023), and **Guanaco-65B** (Dettmers et al., 2023). These models vary in size from 7B to 65B, offering a broad spectrum for performance evaluation and comparison. Please refer to Appendix A for a comprehensive overview of benchmark scores for these models.

We use ChatGPT as a control to evaluate the performance of each model (pre and post-refinement), and GPT-4 as our oracle, due to its exceptional reasoning capabilities (OpenAI, 2023). Please refer to Appendix C for the specific prompts used.