# OpenReview forum: "Democratizing LLMs: An Exploration of Cost-Performance Trade-offs in Self-Refined Open-Source Models"
_EMNLP/2023/Conference — EMNLP 2023 Findings_

### Official Review · Reviewer_27dr · 2023-07-31

**Soundness:** 3

**Excitement:**

3: Ambivalent: It has merits (e.g., it reports state-of-the-art results, the idea is nice), but there are key weaknesses (e.g., it describes incremental work), and it can significantly benefit from another round of revision. However, I won't object to accepting it if my co-reviewers champion it.

**Paper Topic And Main Contributions:**

It is an important task to make open source LLM on par with proprietary models. This paper proposes a domain-agnostic self-refinement approach to enhance the performance of open-source Large Language Models (LLMs). The authors also introduce a novel ranking metric called Performance, Refinement, and Inference Cost Score (PeRFICS) to evaluate and compare the performance and cost-effectiveness of different models. The experimental results demonstrate significant performance improvements achieved through self-refinement, with an average improvement of 8.2% across various models. The PeRFICS metric provides a comprehensive evaluation of models, taking into account their baseline performance, refinement improvement, and inference cost. The paper also presents three case studies in email response automation, video game non-player character (NPC) AIs, and corporate LLM deployment, showcasing the applicability and benefits of PeRFICS in different scenarios.

Overall, the paper presents an innovative approach and a comprehensive ranking metric for enhancing the performance of open-source LLMs. The experimental results demonstrate the effectiveness of the proposed self-refinement approach, and the case studies illustrate its practical applicability. However, there are some limitations and areas for improvement, such as providing more implementation details.

**Questions For The Authors:**

In line 931, you mentioned "Question". However, I don't see any of that. Do I miss anything?

**Reasons To Accept:**

1. Innovative approach: The domain-agnostic self-refinement approach proposed in this paper provides a new method to enhance the performance of open-source LLMs without relying on external feedback or pre-training.
2. Comprehensive ranking metric: The PeRFICS metric takes into account multiple factors, including baseline performance, refinement improvement, and inference cost, providing a balanced evaluation of different models.
3. Experimental validation: The experimental results demonstrate significant performance improvements achieved through self-refinement across various models of different sizes. The case studies further illustrate the practical applicability and benefits of PeRFICS in different scenarios.

**Reasons To Reject:**

My main concern is the limited implementation details: The paper does not provide sufficient implementation details for reproducing the study. It might be useful if more details are provided in section 3.1.1. I find it a little bit hard to figure out the exact procedure of domain-agnostic self-refinement after reading 3.1.1.

**Reproducibility:**

4: Could mostly reproduce the results, but there may be some variation because of sample variance or minor variations in their interpretation of the protocol or method.

**Reviewer Confidence:**

1: Not my area, or paper was hard for me to understand. My evaluation is just an educated guess.

---

> ### Author Rebuttal · Authors · 2023-08-28
>
> Dear reviewer 27dr,
>
> Thank you for your thoughtful review and recognition of the innovative aspects of our work. We appreciate the opportunity to address your concerns.
>
>
> ---
>
>
> ### **Implementation Details in Section 3.1.1**
>
>
>
> - We understand that your primary concern lies in implementation details outlined in Section 3.1.1. The goal of the aforementioned section is to provide an algorithmic background into our domain-agnostic refinement approach, and therefore, in order to not detract from the primary focus, we decided to include the relevant fine-grained implementation details in Appendix C and Appendix D.
>
> - Here, we detail the exact prompts used and the model responses. We also specify the hyperparameters used for token generation and the tunable parameters for our model ranking according to the PeRFICS metric.
>
>
>
> However, we recognize your concerns, and will ensure that we add additional signposts referencing the appendix (similar to the one on line 386).
>
>
>
> **Do you think that adding more signposts and references to the relevant implementation details would address your concerns effectively?**
>
>
>
> ---
>
>
>
> ### **Clarification on Line 931**
>
>
>
> - As detailed in Section 3.1.1, after instruction _I_zero_, we pass in the necessary row of our benchmark dataset, which is a question.
>
> - Therefore, in order to prime the model (so that it knows what to expect and can answer the question effectively), we have the string “Question:” as our token leading to the appended question. We hope this clarifies your concern regarding line 931's open “Question:”.
>
>
>
>
> **We'll ensure clearer exposition of this in our revised manuscript. We appreciate your attention to detail and hope this addresses your concern.**
>
>
>
> ---
>
>
>
> ### **Reproducibility Concerns**
>
>
>
> - The questions used, the model responses, the oracle’s evaluation and the code necessary to generate the same are provided in the additional supplementary materials.
>
> - We do not use any external, non-public datasets in effort of maintaining the highest standards of reproducibility.
> - We will also move some of the implementation details from the appendix back to the main section to aid this effort.
>
>
>
> **Could you provide us with some specific areas within our codebase or implementation details where you have reproducibility concerns? We would be glad to clarify or improve accordingly.**
>
>
>
> ---
>
>
>
> _Please do let us know if you have any further questions or concerns regarding the soundness or reproducibility of our work, and we shall be more than happy to address them._
>
>
>
> ---
>
>
>
> Thank you once again for your insightful comments.

---

### Official Review · Reviewer_7UkX · 2023-08-02

**Soundness:** 2

**Excitement:**

3: Ambivalent: It has merits (e.g., it reports state-of-the-art results, the idea is nice), but there are key weaknesses (e.g., it describes incremental work), and it can significantly benefit from another round of revision. However, I won't object to accepting it if my co-reviewers champion it.

**Paper Topic And Main Contributions:**

- This paper delves into an investigation of cost-performance compromises in autonomous improvement of open-source large language models (LLMs).
- The authors propose a domain-agnostic self-enhancement method that evades external interferences, along with a new evaluation measure named PeRFICS, which strives to identify the best model for a given task, keeping in mind both performance and cost.
- Empirical findings from the Vicuna benchmark reveal that this domain-neutral self-enhancement method can assist open-source LLMs in enhancing the quality of their outputs and exceed ChatGPT in performance across most task domains. Additionally, this paper provides three case studies illustrating how effectively PeRFICS can guide users in selecting the most suitable model.


**Reasons To Accept:**

This paper addresses a valuable question regarding the self-refinement of open-source large language models (LLMs).

**Reasons To Reject:**

- It appears that this paper merely employs a simplified version of the Self-Refine method (https://arxiv.org/abs/2303.17651) initially devised for GPT, and applies it to open-source LLMs, without presenting any substantial differentiation.
- Recent research indicates a considerable discrepancy between the auto-evaluation methods based on GPT-4 and human evaluations, thus casting doubt on the reliability of the evaluation method used in this paper and the persuasiveness of its experimental results.
- In Section 5, the authors present Figure 1, Figure 2, and Figure 3, yet fail to provide explanations or analysis for them. The authors need to clarify why there are negative numbers in Figure 1, as well as the implications of Figure 2 and Figure 3.
- This paper's methodology has a strong link to Self-Refine, but it lacks results from Self-Refine on the Vicuna benchmark in Section 5.


**Reproducibility:**

4: Could mostly reproduce the results, but there may be some variation because of sample variance or minor variations in their interpretation of the protocol or method.

**Reviewer Confidence:**

4: Quite sure. I tried to check the important points carefully. It's unlikely, though conceivable, that I missed something that should affect my ratings.

---

> ### Author Rebuttal · Authors · 2023-08-28
>
> Dear reviewer 7UkX,
>
> Thank you for taking the time to read our work and offering your valuable feedback.
>
>
> ## **Paper Focus and lack of results from self refine**
>
>
>
> - You are accurate in recognizing that our method is a variant of Self Refine [9](https://arxiv.org/abs/2303.17651) (as mentioned in our manuscript - lines 209 - 212). We would like to use this opportunity to elaborate upon our core motivations for this work, which were not only to design this domain agnostic framework, but also to devise a metric (PeRFICS) to rank the “refinability” of SoTA open source models of varying sizes.
>
> ### **How our method differs from self refine**
>
> - Self Refine follows a targeted approach for refinement and critique (Madaan et al, Page 31, Appendix M.1) [9](https://arxiv.org/abs/2303.17651), where up to six in-context examples are passed in to assist with feedback process. This can also be seen in their attached code base on Github [10](https://github.com/madaan/self-refine/blob/main/src/commongen/data.py), where very fine-grained feedback examples are provided to the feedback generator, thereby inducing a bias in the refinement process.
>
> - The ability of models to improve from in-context prompting is an emergent property of larger LLMs (GPT-3 and bigger) [11](https://arxiv.org/abs/2303.03846), as demonstrated by a Google Brain research paper. **This finding is replicated by the Self Refine paper [9](https://arxiv.org/abs/2303.17651), where they explicitly state that they were not able to replicate their experimental results for smaller models, specifically Vicuna 13B (Madaan et. al, Page 7, Section 4 - Does Self Refine Work With Weaker Models)** [9](https://arxiv.org/abs/2303.17651).
>
> - They acknowledge that their experiments failed to produce consistent results for Vicuna 13B. Due to this acknowledgement and high costs associated with repeated experimentation, we decided to not extensively test our method against Self Refine. **We acknowledge this as a limitation of our study, and we will revise our limitations section to reflect this. Thank you for your constructive feedback.**
>
> - However, due to the domain agnostic nature of our refinement process, which does not harness the use of in-context prompts to guide the feedback engine driving the ratings, we are able to achieve SoTA refinement results **without in-context prompting**.
>
> _To reiterate, our core motivation was to demonstrate the consumer applicability of refined open source LLMs (as evidenced by our case studies) which are inherently of smaller sizes, thereby allowing researchers and users to make educated decisions with regard to model selection using PeRFICS._
>
>
> ## **GPT-4 As Our Evaluator**
>
> **TL;DR**: We utilized GPT-4 as an oracle based on its proven reliability in recent literature ([1](https://aclanthology.org/2023.acl-long.870), [2](https://arxiv.org/abs/2306.05685), [3](https://aclanthology.org/2023.bea-1.32.pdf), [4](https://arxiv.org/abs/2308.02575)). While we acknowledged and addressed GPT-4 biases, such as positional preference ([5](https://github.com/lm-sys/FastChat/issues/826)), through methodological adjustments detailed in Appendix E, GPT-4's consistent evaluation behavior minimized the need for multiple runs. Our manual review confirmed GPT-4's accuracy in our context. To strengthen our findings, we propose testing with multiple oracles, including Llama2 and Platypus2. Updates will be incorporated in our revised manuscript.
>
> ---
>
> - We understand your concerns about the untrustworthiness of GPT-4 as a strong evaluator. Our motivation for using GPT-4 as an oracle was influenced by [1](https://aclanthology.org/2023.acl-long.870), where the stability and adaptability of using an LLM evaluator was demonstrated for open ended story generation and adversarial attack tasks.
>
>
>
> - In extremely recent research by the Vicuna team [2](https://arxiv.org/abs/2306.05685), we see that GPT-4, when acting as a judge for tasks on the Vicuna benchmark, achieves over 80% agreement with other humans - which is the same level of agreement between independent human judges.
>
>
>
> - An ACL paper by Duolingo research [3](https://aclanthology.org/2023.bea-1.32.pdf), showcases that GPT-4 can evaluate human written discourse in a reliable, consistent and in a manner that is in strong agreement with human judgment.
>
>
>
> - These results are echoed in other papers such as [4](https://arxiv.org/abs/2308.02575), where GPT-4 is shown to provide reliable ratings in domains of higher education, which speaks to the calibre of GPT-4s evaluation prowess.
>
>
>
> However, this is not to say that GPT-4 is without fault and a perfect evaluator.
>
>
>
> - In our experimentation, we did notice a significant positional, ordering and labeling bias, as referenced in Section 5, Line 438. For eg: GPT-4 tended to prefer the first input grading candidate. This issue was later brought up by an independent researcher in Vicuna team's codebase [5](https://github.com/lm-sys/FastChat/issues/826), and was acknowledged by the first author.
>
>
>
> - However, this did not impact our research as we already had taken the appropriate steps to mitigate these biases, such as running the experiment multiple times and creating unique pairwise orderings, which were then averaged to mitigate the biases. Appendix E, line 1105 onwards, showcases the base data, pre-aggregation.
>
>
>
> ---
>
>
>
> ### **Manual Review**
>
> In order to leave no room for doubt, the authors and a team of researchers have also manually hand-reviewed the responses, the critiques and the refined improvements of each model, to assess the consistency and accuracy of GPT-4, and have reached a general consensus, regarding the overall capability of GPT-4 as an evaluator in this particular use case.
>
>
>
>
> **We will revise our paper's discussion section to address the same.**
>
> ---
>
>
> ### **Additional Experiment Proposal**
>
>
>
> To further solidify the basis of our work, **we propose the following additional experiment** - where we evaluate the responses with multiple oracles - including, but not limited to Meta’s new Llama2 [6](https://arxiv.org/abs/2307.09288), Platypus2 [7](https://arxiv.org/abs/2308.07317) and other SoTA LLMs that have demonstrated strong performance on the HuggingFace OpenLLM leaderboard [8](https://huggingface.co/spaces/HuggingFaceH4/open_llm_leaderboard). Please let us know if this approach with an additional experiment would aid in solidifying the results of our research.
>
> ---
>
> ## **Figure Signposts and Formatting**
>
> Thank you for pointing out the lack of in-text references and explanations for the figures.
>
> - Figure 1 has negative numbers, as in our experimentation, smaller models have a tendency to be more terse in their responses during the second iteration - post critique. We have explained this phenomenon (Lines 493 - 497), but, as you rightfully have pointed out, we have not explicitly explained the negative values in the image, nor added a linked reference to the same.
>
> - While we have drawn upon inferential statistics in our discussion from Figures 2 and 3, we seem to have not referenced them explicitly.
>
> Thank you for your keen observation and notice of this oversight. We will be sure to add explanations for the negative values in Figure 1, and relevant linked references for the figures 2 and 3, where we explain observed phenomena.
>
>
> ---
>
> ## **References**
>
>
>
> [1] Chiang, C.-H. et al. (2023) Can Large Language Models Be an Alternative to Human Evaluations?, Proceedings of the 61st Annual Meeting of the Association for Computational Linguistics (Volume 1: Long Papers). Available at: https://aclanthology.org/2023.acl-long.870 (Accessed: 25 August 2023).
>
>
>
> [2] Zheng, L. _et al._ (2023) _Judging LLM-as-a-judge with MT-bench and Chatbot Arena_, _arXiv.org_. Available at: https://arxiv.org/abs/2306.05685 (Accessed: 25 August 2023).
>
>
>
> [3] Naismith, B. _et al._ (2023) _Automated evaluation of written discourse coherence using GPT-4_. Available at: https://aclanthology.org/2023.bea-1.32.pdf (Accessed: 25 August 2023).
>
>
>
> [4] Hackl, V. _et al._ (2023) _Is GPT-4 a reliable rater? evaluating consistency in GPT-4 text ratings_, _arXiv.org_. Available at: https://arxiv.org/abs/2308.02575 (Accessed: 25 August 2023).
>
>
>
> [5] Lm-Sys (2023) _Evidence of bias in the ‘fun’ evaluation method using GPT-4 scores? · issue #826 · LM-Sys/FastChat_, _GitHub_. Available at: https://github.com/lm-sys/FastChat/issues/826 (Accessed: 25 August 2023).
>
>
>
> [6] Touvron, H. et al. (2023) Llama 2: Open Foundation and fine-tuned chat models, arXiv.org. Available at: https://arxiv.org/abs/2307.09288 (Accessed: 28 August 2023).
>
>
>
> [7] Lee, A.N., Hunter, C.J. and Ruiz, N. (2023) Platypus: Quick, cheap, and powerful refinement of llms, arXiv.org. Available at: https://arxiv.org/abs/2308.07317 (Accessed: 28 August 2023).
>
>
>
> [8] Open LLM leaderboard - a hugging face space by huggingfaceh4 (2023) Open LLM Leaderboard - a Hugging Face Space by HuggingFaceH4. Available at: https://huggingface.co/spaces/HuggingFaceH4/open_llm_leaderboard (Accessed: 28 August 2023).
>
>
> [9] Madaan, A. _et al._ (2023) _Self-refine: Iterative refinement with self-feedback_, _arXiv.org_. Available at: https://arxiv.org/abs/2303.17651 (Accessed: 25 August 2023).
>
> [10] Madaan, A. (2023) _Self Refine Code Repository (Github)_, _GitHub_. Available at: https://github.com/madaan/self-refine/blob/main/src/commongen/data.py (Accessed: 25 August 2023).
>
> [11] J. Wei et al., “Larger language models do in-context learning differently,” arXiv.org, https://arxiv.org/abs/2303.03846 (accessed Aug. 28, 2023).

---

### Official Review · Reviewer_9ySn · 2023-08-04

**Soundness:** 4

**Excitement:**

4: Strong: This paper deepens the understanding of some phenomenon or lowers the barriers to an existing research direction.

**Paper Topic And Main Contributions:**

This work points out some issues such as privacy in Large Language Models (LLMs). It addresses these issues by proposing a generalized variant of self-critique and self-refinement without external influence and a novel ranking metric Performance, Refinement, and Inference Cost Score (PeRFICS) to find the optimal model. The proposed methodology shows some impressive performance improvement in experiments.

**Questions For The Authors:**

1. How to justify the trustworthiness of the testing results if GPT-4 is not perfect? Can the authors give any theoretical analysis for the performance guarantee?
2. How does the proposed methodology perform in different experiment runs?

**Reasons To Accept:**

1. The research problem in this work is important.
2. The proposed method and ranking metric are novel.
3. The experiments on benchmark datasets including different tasks, and the proposed work shows promising results.


**Reasons To Reject:**

A few concerns as follows:

1. This work uses GPT-4 as the oracle, but the correctness of the oracle is not guaranteed, which may limit the trustworthiness of the evaluation results.

2. It is suggested to add more preliminaries and examples in introduction to better introduce the background and motivation.

3. To better evaluate the robustness of the proposed methodology, standard deviation or statistical tests for the experimental results are encouraged to be added.


**Reproducibility:**

4: Could mostly reproduce the results, but there may be some variation because of sample variance or minor variations in their interpretation of the protocol or method.

**Reviewer Confidence:**

3: Pretty sure, but there's a chance I missed something. Although I have a good feel for this area in general, I did not carefully check the paper's details, e.g., the math, experimental design, or novelty.

---

> ### Author Rebuttal · Authors · 2023-08-28
>
> Dear reviewer 9ySn,
>
> Thank you for your helpful comments. We are glad you found our research problem important and ranking metric novel.
>
>
> ## **GPT-4 As Our Evaluator**
>
> **TL;DR**: We utilized GPT-4 as an oracle based on its proven reliability in recent literature ([1], [2], [3], [4]). While we acknowledged and addressed GPT-4 biases, such as positional preference ([5]), through methodological adjustments detailed in Appendix E, GPT-4's consistent evaluation behavior minimized the need for multiple runs. Our manual review confirmed GPT-4's accuracy in our context. To strengthen our findings, we propose testing with multiple oracles, including Llama2 and Platypus2. Updates will be incorporated in our revised manuscript.
>
> ---
>
> - We understand your concerns about the untrustworthiness of GPT-4 as a strong evaluator. Our motivation for using GPT-4 as an oracle was influenced by [1](https://aclanthology.org/2023.acl-long.870), where the stability and adaptability of using an LLM evaluator was demonstrated for open ended story generation and adversarial attack tasks.
>
> - In extremely recent research by the Vicuna team [2](https://arxiv.org/abs/2306.05685), we see that GPT-4, when acting as a judge for tasks on the Vicuna benchmark, achieves over 80% agreement with other humans - which is the same level of agreement between independent human judges.
>
> - An ACL paper by Duolingo research [3](https://aclanthology.org/2023.bea-1.32.pdf), showcases that GPT-4 can evaluate human written discourse in a reliable, consistent and in a manner that is in strong agreement with human judgment.
>
> - These results are echoed in other papers such as [4](https://arxiv.org/abs/2308.02575), where GPT-4 is shown to provide reliable ratings in domains of higher education, which speaks to the calibre of GPT-4s evaluation prowess.
>
> However, this is not to say that GPT-4 is without fault and a perfect evaluator.
>
> - In our experimentation, we did notice a significant positional, ordering and labeling bias, as referenced in Section 5, Line 438. For eg: GPT-4 tended to prefer the first input grading candidate. This issue was later brought up by an independent researcher in Vicuna team's codebase [5](https://github.com/lm-sys/FastChat/issues/826), and was acknowledged by the first author.
>
> - However, this did not impact our research as we already had taken the appropriate steps to mitigate these biases, such as running the experiment multiple times and creating unique pairwise orderings, which were then averaged to mitigate the biases. Appendix E, line 1105 onwards, showcases the base data, pre-aggregation.
>
> ---
>
> ### **Evaluation Across Different Runs**
>
> - GPT-4's demonstrated consistency across repetitions, as evidenced in [4](https://arxiv.org/abs/2308.02575). The results in this study suggest that with clear prompts and proper system settings, GPT-4 exhibits deterministic behavior. This consistent output not only ensures reliability on the first pass but also potentially eliminates the need of repetitive runs. Due to the high cost of multiple passes through GPT-4, and cost of correcting for the aforementioned bias, we were unable to perform multiple runs.
>
> **Thank you for bringing this to our attention. We will address this as a limitation in our revised manuscript.**
>
> ---
>
> ### **Manual Review**
> In order to leave no room for doubt, the authors and a team of researchers have also manually hand-reviewed the responses, the critiques and the refined improvements of each model, to assess the consistency and accuracy of GPT-4, and have reached a general consensus, regarding the overall capability of GPT-4 as an evaluator in this particular use case.
>
>
> **We will revise our paper's discussion section to address the same.**
>
> ---
>
> ### **Additional Experiment Proposal**
>
> To further solidify the basis of our work, **we propose the following additional experiment** - where we evaluate the responses with multiple oracles - including, but not limited to Meta’s new Llama2 [6](https://arxiv.org/abs/2307.09288), Platypus2 [7](https://arxiv.org/abs/2308.07317) and other SoTA LLMs that have demonstrated strong performance on the HuggingFace OpenLLM leaderboard [8](https://huggingface.co/spaces/HuggingFaceH4/open_llm_leaderboard). Please let us know if this approach with an additional experiment would aid in solidifying the results of our research.
>
>   ---
>
> ## **Foundational Concepts and Illustrative Examples**
>
> We agree with the reviewer that we can improve upon this submission by adding teaser figures and preliminary analysis in our introduction section. We will include an image which will showcase the zero-shot vs refined response examples, and potentially additional case studies as well.
>
> ---
>
> We greatly appreciate your review. Including these changes will create a stronger submission. Please let us know if you've any additional concerns. Thank you once again for your time and in-depth analysis.
>
> ## **References**
>
> [1] Chiang, C.-H. et al. (2023) Can Large Language Models Be an Alternative to Human Evaluations?, Proceedings of the 61st Annual Meeting of the Association for Computational Linguistics (Volume 1: Long Papers). Available at: https://aclanthology.org/2023.acl-long.870 (Accessed: 25 August 2023).
>
> [2] Zheng, L. _et al._ (2023) _Judging LLM-as-a-judge with MT-bench and Chatbot Arena_, _arXiv.org_. Available at: https://arxiv.org/abs/2306.05685 (Accessed: 25 August 2023).
>
> [3] Naismith, B. _et al._ (2023) _Automated evaluation of written discourse coherence using GPT-4_. Available at: https://aclanthology.org/2023.bea-1.32.pdf (Accessed: 25 August 2023).
>
> [4] Hackl, V. _et al._ (2023) _Is GPT-4 a reliable rater? evaluating consistency in GPT-4 text ratings_, _arXiv.org_. Available at: https://arxiv.org/abs/2308.02575 (Accessed: 25 August 2023).
>
> [5] Lm-Sys (2023) _Evidence of bias in the ‘fun’ evaluation method using GPT-4 scores? · issue #826 · LM-Sys/FastChat_, _GitHub_. Available at: https://github.com/lm-sys/FastChat/issues/826 (Accessed: 25 August 2023).
>
> [6] Touvron, H. et al. (2023) Llama 2: Open Foundation and fine-tuned chat models, arXiv.org. Available at: https://arxiv.org/abs/2307.09288 (Accessed: 28 August 2023).
>
> [7] Lee, A.N., Hunter, C.J. and Ruiz, N. (2023) Platypus: Quick, cheap, and powerful refinement of llms, arXiv.org. Available at: https://arxiv.org/abs/2308.07317 (Accessed: 28 August 2023).
>
> [8] Open LLM leaderboard - a hugging face space by huggingfaceh4 (2023) Open LLM Leaderboard - a Hugging Face Space by HuggingFaceH4. Available at: https://huggingface.co/spaces/HuggingFaceH4/open_llm_leaderboard (Accessed: 28 August 2023).

---

### Meta-Review · Area_Chair_k4nT · 2023-09-18

**Recommendation:** 3

**Metareview:**

This paper proposes a domain-agnostic self-refinement approach with a novel ranking metric to enhance the performance of open-source Large Language Models (LLMs) that evades external interferences. The paper also performs extensive evaluations

Pros:
- The paper is solving an important and relevant problem
- The ranking metrics are novels and comprehensive
- Promising results on benchmark datasets across multiple tasks.
- The authors have provided code

Cons:
- Unclear description of experiment results
- Bias in using GPT-4 as a evaluator
- Empirical comparison to self-refine. The authors have compared to self-refine in introduction and related work but comparison to self-refine in experiments is missing

---

### Decision · Program_Chairs · 2023-10-07

**Decision:**

Accept-Findings

**Comment:**

This paper proposes a domain-agnostic self-refinement approach with a novel ranking metric to enhance the performance of open-source Large Language Models (LLMs) that evades external interferences. The paper also performs extensive evaluations

Pros:
- The paper is solving an important and relevant problem
- The ranking metrics are novels and comprehensive
- Promising results on benchmark datasets across multiple tasks.
- The authors have provided code

Cons:
- Unclear description of experiment results
- Bias in using GPT-4 as a evaluator
- Empirical comparison to self-refine. The authors have compared to self-refine in introduction and related work but comparison to self-refine in experiments is missing